# Intelligent Drone Positioning via BIC Optimization for Maximizing LPWAN Coverage and Capacity in Suburban Amazon Environments

**DOI:** 10.3390/s23136231

**Published:** 2023-07-07

**Authors:** Flávio Henry Cunha da Silva Ferreira, Miércio Cardoso de Alcântara Neto, Fabrício José Brito Barros, Jasmine Priscyla Leite de Araújo

**Affiliations:** Institute of Technology, Federal University of Pará (UFPA), Belém 66075-110, Brazil; henryferreira014@gmail.com (F.H.C.d.S.F.); fbarros@ufpa.br (F.J.B.B.); jasmine@ufpa.br (J.P.L.d.A.)

**Keywords:** wireless sensor networks, IoFT, LoRa, coverage optimization, channel modeling, bioinspired computing

## Abstract

This paper aims to provide a metaheuristic approach to drone array optimization applied to coverage area maximization of wireless communication systems, with unmanned aerial vehicle (UAV) base stations, in the context of suburban, lightly to densely wooded environments present in cities of the Amazon region. For this purpose, a low-power wireless area network (LPWAN) was analyzed and applied. LPWAN are systems designed to work with low data rates but keep, or even enhance, the extensive area coverage provided by high-powered networks. The type of LPWAN chosen is LoRa, which operates at an unlicensed spectrum of 915 MHz and requires users to connect to gateways in order to relay information to a central server; in this case, each drone in the array has a LoRa module installed to serve as a non-fixated gateway. In order to classify and optimize the best positioning for the UAVs in the array, three concomitant bioinspired computing (BIC) methods were chosen: cuckoo search (CS), flower pollination algorithm (FPA), and genetic algorithm (GA). Positioning optimization results are then simulated and presented via MATLAB for a high-range IoT-LoRa network. An empirically adjusted propagation model with measurements carried out on a university campus was developed to obtain a propagation model in forested environments for LoRa spreading factors (SF) of 8, 9, 10, and 11. Finally, a comparison was drawn between drone positioning simulation results for a theoretical propagation model for UAVs and the model found by the measurements.

## 1. Introduction

Wireless communications have an intricate relationship with the technologies of the Internet of Things (IoT), a fairly new concept made for uniting and connecting all sorts of devices, services, and commodities in the contemporary world. Many of such wireless data-transmitting protocols associated with IoT are high-speed networks to favor the heavy data usage of mobile users, such as wide-area networks (WAN) and 4G-LTE or 5G systems. However, it is not always that these high-power, high-data-rate services are needed. Often, devices with low battery consumption cannot bear to operate at such power-consuming paces, which characterizes a need for another kind of network: the low-power wide-area networks (LPWAN).

LPWAN are systems designed to work with low data rates but still keep the extensive area coverage provided by high-powered networks. Thus, a private and secure network for the connection of low-power devices can be established, keeping financial costs and battery consumption at a minimum. For instance, wireless sensor networks, or WSNs, are known to function effectively with models of different types of LPWAN, as seen in [1]. One of such types is long-range, also known as LoRa, which is currently extensively studied and the most readily available.

In [2], the authors discourse over the functioning of LoRa, its benefits, and its challenges. Given that it is a bidirectional manner of sending data between users and transmitters, a LoRaWAN can easily export information such as geolocation, signal-interference-plus-noise ratio (SINR), and received signal strength indicators (RSSI). This kind of flexibility and inherent monitoring of the network makes the integration between receiving sensors and transmitting drones a promising implementation.

As LoRa modules are lightweight and generally have their own separate battery, they can easily be attached to UAVs without interfering with their flight capabilities or power consumption. Given that the battery life of drones is generally low due to their high-power flight demands, LoRa is an economic alternative to make drone-array networks work.

For this paper, the focus is set on how to optimize an array of UAVs for a LoRaWAN network, to give the best coverage area possible, in which all network users/sensors are satisfactorily connected. In order to accomplish this coverage area maximization, some metaheuristic optimization techniques are to be used to minimize the outage probability of sensors. These will also determine which details each sensor will use to connect to the drone gateways so that they can obtain maximum coverage and maximum data rate. Therefore, a simulation was constructed in MATLAB software, simulating a forested, suburban environment, with the purpose of giving results that shall be applicable to the region in which the measured data were collected.

The major contribution that the work herein aims to achieve is the analysis between simulated and measured signal propagation results in light-to-medium forested environments via UAVs, using the campus of the Federal University of Pará (or UFPA) as a reference within the Amazon rainforest area. The optimization technique is then applied to position these UAVs in order to better attend to the needs of users and maximize coverage inside of this context.

The structure of this text is divided into said categories: Section 3 explains the methodology utilized in our studies, divided into three subsections: LoRa Propagation Models, Bioinspired Computing Algorithms, and Drone Positioning Simulations; Section 3 displays results of the simulations for both theoretical and measured propagation models. Section 4 discourses on the meaning and relevance of said results. Finally, Section 5 sums up the paper as its conclusion, as well as providing future study alternatives.

## 2. Related Works

In the literature, there are related works that deal with the objectives proposed herein separately, but it is difficult to find any that tie them all together in a single study. For instance, there are many articles that implement a LoRaWAN with the use of UAVs, as well as numerous ones that propose drone-array optimization via metaheuristics, and some select few that address a propagation model for UAVs in forested areas. Therefore, some studies that have inspired the study presented in this paper are to be explained below.

In [3], a model for a LoRa system of data collecting for precision agriculture is proposed. The aim of this study is to perceive how close must the drone in this type of setting fly by the sensors in order to collect data within a given quality threshold. It considers the pathing of only one drone in a plantation grid that has various sensors separated from each other by a certain distance k. Some metrics are considered in the study for both small and large plantation fields, such as the minimum autonomy time for the drone, the packet transmission times for different SF, and the maximum number of sensors that a drone can attend for different SF.

The authors of [4] present a rich survey about the usage of UAVs in real-time LoRa applications. Not only does it contain much of the technical and theoretical information about its bit rate, spreading factor, transmission power, coding rate, bandwidth, and carrier frequencies around the world, but a plethora of papers showing UAV-based modules and UAV-based gateways are also evident within the survey. Some of the most interesting are [5], which focuses on the detection of gas leakages with air and humidity sensors, with a camera feed enhanced via machine learning; [6] in which a wireless sensor network is proposed for marine environmental monitoring, with compelling real-life results; and [7], a study on adaptable usage between LoRa and Wi-Fi for high-data management in agricultural applications.

Many articles are available in the literature about the employment of metaheuristic computation and machine learning in order to coordinate or facilitate the control, pathing, and positioning of drone arrays. One of such is found in [8], whence optimization methods such as the genetic algorithm (GA) and simulated annealing (SA) with the intent of better positioning UAV base stations (UAV-BS) propagating in the 5G spectrum. The results of the simulation are then analyzed for an area of 80 square kilometers, and the objective function to be minimized is the difference between the total area and the integration of areas covered by each of the UAVs.

In [9], however, a heuristic algorithm was proposed to function on the optimization of drone positioning in a 3D environment in order to supply a good area of coverage and data rates for densely populated user regions. The chosen optimization method in this paper is particle swarm optimization (PSO), and due to its inherent velocity, the results are very promising. The algorithm can also determine whether the number of drones is satisfactory or redundant, as well as the maximum data rate of users which can be served by a UAV.

It seems that many of the works found in the literature for drone-array coordination and positioning are, in fact, based on PSO and its variants. For instance, in [10], PSO is utilized to perform a simpler task of finding an optimal position in a 3D space for UAVs in emergency network situations. Similar results are attested in [11], in which UAVs are deployed for disaster management situations. However, in [12], a cooperative search algorithm (based on PSO) optimizes the density and distance between micro-UAVs in a swarm-based micro-drone system. The objective of the study is to prevent UAV clustering in heavily occupied areas, thus distributing some of the drones to underpopulated areas within the search space, potentially avoiding local optima and widening the system’s coverage area.

Some surveys on the usage of machine learning for UAV positioning and signal propagation, which are interesting to denote, are contained in [13,14]. The former contains a plethora of pertinent information about utilizing UAV arrays in communication systems in general, but also draws attention to many articles that deal with trajectory optimization, cellular network planning, and channel modeling via machine learning solutions. The latter, thus, exposes papers that prefer to use a swarm-based learning approach. It also contains studies that analyze drone applications in IoT and that aim to optimize the area of coverage of signals, which are both related to the work made in this paper.

### Contributions

Given all the information provided above on works related to the studies contained in this paper, our contributions can be compared to the rest of the literature. No papers were found on the propagation of LoRa, or LPWANs in general, in forested environments. Additionally, no study has conciliated the usage of UAV arrays and their positioning optimizations with channel modeling in forested environments. Finally, there are not many studies on the usage of UAVs for signal propagation in Amazon rainforest regions, let alone on drone positioning and coverage optimization. This paper aims to fill these necessities and to enrich the literature about LoRa propagation and wireless sensor network optimization in forested areas.

Table 1 denotes the contributions of this work compared to others cited throughout this section.

## 3. Methodology

To better expose the methods applied to our studies, it is important to divide them into three subsections. The first part shall deal with the propagation models that were used as a basis for the simulation of UAV positioning. The second part is about the three bioinspired computing algorithms that were chosen to perform the optimizations. The last part, therefore, explains how the drone-array positioning simulations are generated and conducted via MATLAB and the coding behind them.

### 3.1. Classical Propagation Model for UAV Base Stations

Herein lie the mathematical equations of propagation modeling of wireless signals utilized in the algorithms. A strong theoretical basis for these is of great importance, as it signifies, in the optimization process, the capacity of the algorithms to perceive a user as connected or not, and to which drone should it connect, based on the measurements of received power ($P_{r}$, as is most referred to in the text) and signal-to-interference-plus-noise ratio (SINR).

The trigonometric equations on the positioning of drones are found below in (Equation 1) and (Equation 2). Figure 1 is used to represent the trigonometric variables visually. These equations are found in the work of [15], which is, overall, one of the most utilized propagation models for UAV systems in the literature, as well as in other articles by the same author [16,17,18,19,20].
(1)$$d=\sqrt{R^{2}+h^{2}}$$
(2)$$\theta =\arctan\frac{h}{R}$$

In which R is the distance from the projection of the drone on the user plane to the user itself, h is the UAV height in relation to the user plane, d is the actual distance from user to UAV, and θ is the elevation angle of the UAV in relation to the user.

So, a theoretical model for drone signal propagation is described below, again inspired by the works in [15]:

The model conveys both line-of-sight (LoS) and non-line-of-sight (NLoS) losses and is represented in Equations (Equation 3) and (Equation 4), and the probability of having a LoS connection for an elevation angle of θ is given by (Equation 5).
(3)$$PL_{LoS(dB)}=20\log\left(\frac{4\pi fd}{c}\right)+\zeta_{LoS},$$
(4)$$PL_{NLoS(dB)}=20\log\left(\frac{4\pi fd}{c}\right)+\zeta_{NLoS},$$
(5)$$P_{LOS}=\frac{1}{Z},Z=1+\alpha \exp\left(-\beta \left[\frac{180}{\pi }\theta \right]-\alpha \right)$$

In which f is the propagated frequency, and $\zeta_{LoS}$ and $\zeta_{NLoS}$ are loss constants related to LoS and NLoS propagations. Furthermore, α and β are environmental constants that are necessary to adapt this model to urban, suburban, or rural ambiences, and $P_{NLOS}=1-P_{LoS}$. Values of d and θ are thus according to (Equation 1) and (Equation 2).

Hence, the average path loss between LoS and NLoS situations is described as:(6)$$PL_{avg(R,h)}=P_{LoS}\times PL_{LoS}+P_{NLoS}\times PL_{NLoS},$$
which is then, by applying Equations (Equation 3)–(Equation 6), transformed into:(7)$$PL_{avg}=20\log\left(\frac{4\pi fd}{c}\right)+\frac{\left(\zeta_{LoS}+Z\zeta_{NLoS}\right)}{\left(1+Z\right)}$$

So, considering transmitting and receiving antenna gain, (Equation 8) denotes the received power of the user based on the path loss:(8)$$P_{r(dB)}=P_{t}+G_{t}+G_{r}-PL_{avg}$$

It is important to denote this is a value unique for each user connected to each drone. In light of this, received power between drones in an array system might cause UAV-to-UAV interference and, as such, has to be considered in the model. In that manner, the SINR metric comes into activity, which determines how many users are connected to the UAV array and the quality of its signal. As said previously, SINR is crucial for the algorithm to decide which UAV should connect to which user, and its implementation is given by:(9)$$SINR_{i,j}=\frac{Pr_{i,j}}{q+\sum_{k=1,k\neq j}^{N_{UAV}}Pr_{i,j}},$$
where received power values for the j-th drone and the i-th user are then transformed into SINR by dividing it for the noise floor level q added to the average of the sum of the received powers of the other drones (that is, the UAV-to-UAV interference). The formula was derived in accordance with SINR information in [20,21]. To calculate this formula more easily, SINR and Pr values should be expressed linearly.

The value of the noise floor q, in our calculations, is the sensitivity value of the chosen spreading factor (SF) of 10, which results in $q_{\left(SF10\right)}$ = −132 dBm [22].

In theory, higher SF values provide a greater coverage area and robustness to noise but lower bit rates, and lower values enhance the bit rate of the signal but sacrifice some areas of operation.

Table 2 refers to the signal sensitivity range and its respective SF values for a bandwidth of 125 KHz and a carrier frequency of 915 MHz, which are the recommended LoRa operation values for Europe and Brazil. The FSK mode (short for frequency shift-keying) specified here is another type of modulation that is also present in LoRa gateways as an alternative to CSS, albeit generally less effective, and it is only listed as means of comparison.

#### Empirical Propagation Models for LoRa in Forested Environments

Measured data to better represent medium to densely forested environments were acquired at the Federal University of Pará (UFPA) in a series of measurement campaigns conducted by the staff of LCT-UFPA (in Portuguese: Laboratório de Computação e Telecomunicações).

An 8-user LoRa gateway module configured by an Arduino UNO was used, as well as 2 compact omnidirectional antennas with a maximum transmitting power sensitivity of around 10 dBm and 1 drone, to which we attached the gateway, to realize measurements at different heights.

The drone was kept at a fixed place, thus emulating its usage as an LPWAN gateway transmitting data (Tx), as it transmitted down to the receiver antenna (Rx), which varied its position. The Rx was attached to a car, and thus further distancing away from the UAV in order to measure the variation of RSSI and SINR values according to distance and to the LoRa mode in which the gateway was configured.

Figure 2 displays the path taken by the car for all measurement campaigns. The journey encompassed a traveling distance of around 2.9 km, sprawling from the UFPA bus garage (in red) all the way to CEAMAZON (in green). The path chosen is painted in blue, and the position of the drone, fixed in all campaigns, is represented as a purple circle in the picture. Often, measurements would be halted exactly halfway in order to exchange the drone’s battery packs and then resume normally. A photo showing the Rx antenna attached to the car and the utilized drone in mid-air is represented in Figure 3. Notice that there is a blue 3D-printed module on the back of the drone, made to attach it to the LoRa gateway.

SF values of 8, 9, 10, and 11 were selected, as well as heights of 6 m, 24 m, 42 m, and 60 m above ground. This amounts for a total of 16 measurement journeys and a total of 2824 points measured.

For the path loss models based on the measured data obtained, the log-distance path loss and close-in free space model were selected (abbreviated to LDPL and CI, respectively), which are widely used as generic models that are able to adapt parameters to an environment of interest [24]. In the work in [25], the authors used some of the different path loss models to compare which would best fit their measured results, the log-distance one being a part of them. Additionally, in the works of [26], the LDPL was used effectively to predict path loss in a densely forested environment. As for CI, there are examples of its usage in LoRa propagation in the works of [27,28].

The equations, thus, were derived from both measured data and the classic LDPL and CI models. The general equations are as follows:(10)$$PL_{LPDL,CI}=PL_{0}+10nlog\left(\frac{d}{d_{0}}\right)+X,$$
(11)$$Pr_{LPDL,CI}=P_{t}+G_{t}+G_{r}-PL,$$
(12)$$PL_{0}=20log\left(\frac{d_{0}}{10^{3}}\right)+20log\left(f\right)+32.5,$$
in which $d_{0}$ is the reference distance (1 m for CI, 10 m for LDPL), $PL_{0}$ is the path loss in the reference distance, n is the path loss exponent (PLE), d is the distance or length of the path, $Pr_{LPDL,CI}$ is an estimate of the received power with zero antenna gains, and X is a normal random variable with mean equaling to zero, which is supposed to emulate the shadowing effects of signal loss. In Equation (Equation 12), the distance is given in meters, and the frequency is given in MHz.

From [20], values for signal amplitude and, therefore, inference of the proper path loss model used to approximate modeled results to measured ones are then transcribed into (Equation 13) and (Equation 14):(13)$$A=10log\frac{d}{do},$$
(14)$$n=A\backslash (P_{t}-RSSI_{\left(measured\right)}+L_{0}),$$
in which A is a distance to reference distance ratio for log-distance and free space models that is to be inputted into the path loss equation. Given that these calculations need to be input in vector or matrix form in MATLAB, there is in (Equation 14) a matrix left division symbol, which is necessary to yield the correct results. Furthermore, $RSSI_{\left(measured\right)}$ is the vector of received power values, or RSSI, featured in the measured data, and $P_{t}$ is the transmitted power. Hence, the path loss exponent is estimated.

Next, for the calculation of the standard deviation σ to be utilized in random variable X, the mean square error between measured results and calculated ones must be employed. So, the equation is as follows
(15)$$MSE=\frac{\sum_{n=1}^{N_{data}}{\left(Pt-RSSI_{\left(measured\right)}-p\right)}^{2}}{N_{data}};,\sigma =\sqrt{MSE}$$
where p is the calculated result of RSSI and $P_{t}$ as in (Equation 11) but without the random variable included. Since MSE is a variance that is considered to be unbiased, it is enough to take its square root to discover the standard deviation.

Since there are different RSSI and distance values than expected in the theoretical calculation, in every set of SF values, path loss exponents may differ for every SF mode, as well as each height. This is because LDPL and CI may vary when taking into consideration the variation of modulation, environment, or Tx/Rx height. Thus, Figure 4a,b represent the LDPL and CI models proposed in specific SF and UAV height conditions. As for Figure 4a, it represents the model for only values in SF 10 and with the drone set at 60 m above the ground as an example of the curve fitting of measured data, whereas Figure 4b displays a calculation over data from SF 11 at a height of 24 m. Values in red asterisks represent the measured data, whilst black asterisks show the LDPL-calculated data with shadowing included. The blue, magenta, and green colors are, respectively, the fittings of the LDPL, CI, and theoretical (Mozaffari) models.

In Table 3 and Table 4, values of path loss exponent, standard deviation, reference loss, and distance found for each SF and height are discoursed. Reference values were set to a distance of 10 m, upon which the reference loss is calculated. A total of 16 different values are then derived for each case of SF and height variation, according to the parameters of path loss exponent (PLE) and the standard deviation of data (σ). The reference loss at d = 10 m is equal to $PL_{o}=51.73$, and d = 1 m is $PL_{CI}=31.73$ in accordance with Equation (Equation 12).

The curve fitting with the values of Table 3 achieves satisfactory results in relation to measured data in each case. It can be denoted that values representing a height of 6 m have greater path loss exponents and a greater dispersion, and this may indicate that the UAV is flying too low to provide good signal coverage. Inside each SF, it is evident in the table that greater heights produce lower path loss, as it also decreases the PLE, at least in the observable interval.

Lastly, since the SINR is an equation that is independent of any path loss model, (Equation 9) is still applied in this case.

### 3.2. Bioinspired Computing Algorithms

Bioinspired computational techniques are based on natural selection and are valid optimization methods for mathematical and engineering applications, where metaheuristics (trial and error) can be deployed to simplify the calculation of complex problems [29].

In the study herein, three of those methods were chosen, which are the cuckoo search (CS), the flower pollination algorithm (FPA), and a tournament-based genetic algorithm (GA). Due to their greater facility in dealing with non-linear, multi-variable problems [30], CS and FPA are utilized here as alternatives to GA, and all techniques are to be compared to observe which one provides the best solutions.

CS and FPA were designed by Xin-She Yang, with CS being the first to be released in 2009, followed by FPA in 2012. However, genetic algorithms span a greater period of existence, research, and usage. It is denoted that the first formulations of this technique are as old as 1972, with a boom in usage around 20 years later in 1992 [31].

#### 3.2.1. Cuckoo Search

The cuckoo search optimization (CS) algorithm, introduced by Yang and Deb in 2009, has demonstrated its effectiveness as a metaheuristic algorithm for various applications in mathematics, industry, and engineering [32]. Inspired by the parasitic behavior of cuckoo birds, which lay their eggs in the nests of other bird species, CS simulates this natural phenomenon as a computational model. Host birds often do not recognize the foreign egg, either ignoring it or abandoning the nest altogether, allowing the cuckoo egg to hatch and grow much faster than the host’s eggs, resulting in the cuckoo chick expelling the other eggs from the nest and gaining access to more food [33]. This behavior serves as the primary inspiration for the CS algorithm, where the cuckoo eggs represent candidate solutions, and the host bird nests are scattered around the search space [32].

According to [33,34], the synthesis of the cuckoo search algorithm can be summarized in three fundamental rules:Each cuckoo lays one egg at a time, to be deposited in a random nest. Every egg is to be considered a unique solution to the problem;The best solutions (eggs) will be carried over through the next iterations, in accordance with the parasitic and survivalist nature of the cuckoo chicks;The number of available nests per iteration is fixed by the developer of the code. The probability of a cuckoo egg being discovered by the host bird is defined as Pa∈[0,1). Then, a discard probability will define if the host bird will get rid of a cuckoo nest or let it hatch.

In CS, cuckoo birds move using Lévy flights, which are random flight paths that each cuckoo (in a group of i cuckoo birds) takes to find nests [33]. To implement these flights, Equation (Equation 16) represents the mathematical expression for Lévy flight, and Equation (Equation 17) represents the mathematical expression for the Lévy distribution [33]. These equations can be translated into code for practical use in the CS algorithm.
(16)$$X_{i}^{t+1}=X_{i}^{t}+\alpha ⊕Levy\left(\beta \right)$$
(17)$$Levy\left(\beta \right)≅u=t^{-(\beta +1)};\left(1<\beta \leq 3\right)$$

In which t is the current iteration, and i is the maximum number of cuckoo birds in the current generation. The step size, represented by the constant α, is adjustable according to the developer’s needs, but must always be greater than zero. In this study, the value of α is set to 1 [33].

In Equation (Equation 16), the Lévy flight is associated with the Lévy distribution using the entrywise multiplication product ⊕, which allows for better utilization of the search space [33]. This is akin to the approach of that used in the particle swarm optimization (PSO) algorithm, which utilizes a similar product.

Equation (Equation 17) pertains to the Lévy distribution, which has infinite variance and average values. The variable β represents the random step length necessary to provide variable magnitudes to the random walk in the Lévy Flight method [33].

A pseudocode of the CS algorithm can be found in the article in which it was proposed by Yang and Deb [33].

#### 3.2.2. Flower Pollination Algorithm

The flower pollination algorithm (FPA) optimization method is inspired by the pollination behavior of natural flowers and incorporates the Lévy flight approach for optimal space search, similar to the cuckoo search algorithm. Empirical evaluations have confirmed that the FPA optimization method may be more efficient than genetic and particle swarm optimizations in both single- and multi-objective applications [35].

For the FPA, the authors cite four guiding rules [35]:Both biotic and cross-pollination are considered a global pollination process, and are performed by pollinators carrying pollen and executing Lévy flights;In contrast, abiotic and self-pollination are considered as local pollination;Flower constancy is then taken as a measure of the probability of reproduction, which is directly proportional to the similarity between two flowers involved in the pollination process;To control both local and global pollination activities, a switch probability p with values from 0 to 1 is utilized. Local pollination can make up a significant fraction p in the overall pollination process due to physical proximity and other environmental factors such as wind.

Two types of pollination are considered: global and local pollination. This prevents the algorithm from getting trapped in local solutions, and instead orients it to discover global solutions to the objective function. To simplify the implementation of the algorithm, it assumes that each plant has only one flower and can pollinate one other flower at a time, whereas in reality, plants may have multiple flowers and millions of pollinating gametes. This simplification allows the plant, flower, and pollinating gamete to be treated as a single solution unit in the FPA optimization method.

The first rule (global pollination) and third rule (flower constancy) of FPA are resumed in mathematical form at (Equation 18):(18)$$X_{i}^{t+1}=X_{i}^{t}+L\left(X_{i}^{t}-g*\right)$$
where $X_{i}^{t}$ is the solution vector $X_{i}$ at iteration t, representing the pollen of number i, and g* is the best solution of the current iteration.

Lévy flights are also used in this optimization method, and are regulated by the pollination strength L, whereby the step size of each flight is determined. In this algorithm, the flights represent the paths taken by insects and pollinator animals within the global search space of the optimization. However, the equation utilized in the FPA algorithm differs from the one employed in the cuckoo search, as it relies on the Mantegna algorithm instead. Essentially, this technique generates pseudo-random step sizes by utilizing normal distributions to ensure optimal performance while conforming to the requirements of the Lévy distribution. Pollination strength L is thus defined as in (Equation 19):(19)$$L\approx \frac{\lambda \Gamma (\lambda )\sin(\pi \lambda /2)}{\pi s^{1+\lambda }},$$
in which Γ(λ) is the standard, classic gamma function found in Lévy flights and other probabilistic and complex number applications.

The Mantegna step size algorithm is shown in (Equation 20):(20)$$s=\frac{U}{{\left|V\right|}^{\frac{1}{\lambda }}},$$
in which *s* is the step size, U represents a Gaussian distribution of variance $\sigma^{2}$, and V also represents a Gaussian distribution but with unitary variance, as can be verified in [36]. In most cases, the lambda can be treated as a constant value between λ∈[0.5,1.5]. The variance when λ=1 also equals 1, simplifying the calculation of this formula.

To implement Rule 2, which involves local pollination, the flower pollination lgorithm (FPA) simulates flower constancy by limiting pollination to a small neighborhood surrounding the reproductive flower’s location. This process can be expressed as:(21)$$X_{i}^{t+1}=X_{i}^{t}+\epsilon \left(X_{j}^{t}-X_{k}^{t}\right),$$

In the above equation, $X_{j}^{t}$ and $X_{k}^{t}$ denote the pollens from different flowers of the same plant species, and ϵ is a scalar factor that controls the step size of the pollination process.

The fourth rule is a probabilistic mechanism that switches between global and local pollination. By adjusting the probability parameter *p* parametrically, the optimization performance can be improved to better suit the requirements of the objective function.

All stages of the algorithm are represented in pseudocode form by the recommendations in [35]. Some details previously discussed can be noticed, such as an if/else switch for global and local pollination, which are found in Lévy flights and random selection, respectively.

#### 3.2.3. Genetic Algorithm

Genetic algorithms (GAs) are a type of computational method inspired by the process of natural selection and genetics. They are used to optimize complex problems in various fields, including engineering, economics, and biology. GAs employ a population-based approach to problem-solving, where a set of possible solutions, called individuals, are evolved over several generations using selection, crossover, and mutation operators. It is known they are employed in many IoT applications in the literature [37,38,39].

The basic principle of GAs is based on Darwin’s theory of natural selection, which states that the fittest individuals are more likely to survive and reproduce. GAs simulate this process by generating a population of candidate solutions, evaluating them based on a fitness function, and selecting the best individuals for further processing. The individuals that are selected undergo genetic operations, such as crossover and mutation, to produce offspring that inherit the characteristics of their parents. The process is repeated over several generations until a satisfactory solution is found.

One of the advantages of GAs is that they can find optimal or near-optimal solutions in a relatively short amount of time, even for problems with a large number of variables or complex constraints. Additionally, GAs can handle noisy or incomplete data, making them useful in real-world applications where data is often imperfect.

There are a few different methods of conducting the selection to perform the crossover operation, such as roulette, Boltzmann, rank, or tournament selection [40]. In this study, a tournament-based selection with single-point crossover was chosen. In summary, tournament selection is a widely used selection method that has shown promising results in various optimization problems. Its ability to control selection pressure and reduce premature convergence makes it a popular choice among researchers and practitioners. However, the performance of tournament selection is dependent on several factors, such as the tournament size and the selection method used within the tournament.

The pseudocode for the tournament-based genetic algorithm used in this can be found in [41]. It is a compact, but not less effective, form of GA with simple computation and fast running time.

### 3.3. Drone Positioning Simulations and Optimization Metrics

In this section, further details are given about the algorithms used for simulating a space search of users and the positioning of drones in order to reach maximum network coverage. Please notice that there are three versions of the simulation, each for all three of the bioinspired algorithms.

Since the utilized LoRa gateway of the measurement campaigns only supports eight concomitant users, then user equipment (UE) association to UAVs is built upon the availability of eight channels per drone. Then, the association process gives priority to sensors that possess the best received power ($P_{r}$) and are closest to the current position of a drone. If the UAV station that the sensor is trying to connect is already at maximum capacity, then another association calculation is performed to find the next best vacant drone.

The next step is to calculate the SINR of said UE, which is achieved by employing Equation (Equation 9). It is known in the literature that a tolerable value for SINR in LoRa networks is around −20 dBm [42,43]. Therefore, if the UE has an SINR below this value, considering all other interfering UAVs and the noise floor, then it will not count as being associated with the network. Along its iterations, the BIC methods are able to identify these non-associated sensors and thus provide better drone positioning optimization.

Another metric that is vital to the optimization process is the outage probability, which is the probability that a given UE might become out of service taking into consideration its received power and the sensitivity of the receiver and should thus be minimized. Adapted from Rappaport [44] and Goldsmith [20], it is calculated as shown in (Equation 22):(22)$$P_{out}\left[P_{r(i,j)}\leq P_{SF}\right]=Q\left(\frac{P_{t}-PL_{\left(i,j\right)}-P_{SF}}{\sigma_{SF}}\right),$$
in which $P_{SF}$ is the sensitivity of the respective chosen SF as in Table 2, $P_{t}$ is the transmitted power, $PL_{\left(i,j\right)}$ is the path loss (with shadowing considered) for a UE of index i connected to a UAV of index j, and σ is the standard deviation of measured data, according to each SF. The function *Q* is the tail probability of a Gaussian distribution, which can be simply calculated in MATLAB with the command *qfunc*.

Other metrics that are not taken into account in the objective function but are analyzed after the results of the simulation are the spectral efficiency (SE) of the sensors and the cell radius (CR) of the UAV gateways due to their importance to network analysis in the literature on sensor networks, e.g., [45,46]. In this paper, the SE is calculated as an average of all UEs in the simulation, and is given by (Equation 23):(23)$$SE_{\left(avg\right)}=\eta D_{c}\log_{2}\left(1+SINR\right),$$
where η is the channel efficiency, $D_{c}$ is the duty cycle of the LoRa network, and the SINR in this formula should be calculated in linear values. The values η=0.7 and $D_{c}=1\%$ were chosen for the simulations, following their usage in [43,45]. Additionally, the cell radius is calculated as the distance in which an outage of 1% is found, in meters. Following Equation (Equation 22) for $P_{out}=1\%$, CR formulas are deducted for the theoretical path loss of Mozaffari et al. (Equation 6) and for our empirical LDPL and CI path loss models (Equation 10) in order to compare the coverage given by both. Those are displayed below in Equations (Equation 24) and (Equation 25), respectively.
(24)$$CR_{classical}=\frac{c}{4\pi f}\cdot 10^{\frac{P_{t}-P_{SF}-2.325\sigma }{20}-\left(\frac{\zeta_{LoS}+Z\zeta_{NLoS}}{20(1+Z)}\right)}$$
(25)$$CR_{empirical}=d_{0}\cdot 10^{\left(\frac{P_{t}-2.325\sigma -P_{SF}-PL_{0}}{10n}\right)}$$

So, the objective function for all optimizers is given in (Equation 26). It considers two different objectives: average outage probability and UAV requirements. The former is the mean value of the outage defined in (Equation 22) for all UE (in %), and the latter is a measurement of the minimum number of UAVs necessary to meet user association requirements, defined in (Equation 27).
(26)$$Z=0.9\cdot P_{out(avg)}+0.1\cdot UAV_{req}$$
(27)$$UAV_{req}=\left(UAV-1\right)+N_{Users}-N_{Associated},$$
in which UAV is the optimal quantity of UAVs given as input by the optimizers, $N_{U}sers$ is the total number of UEs in the simulation, and $N_{Associated}$ is the number of UEs considered to be covered by the network and not experiencing an outage. The variable $N_{Associated}$ also implies that these associated users already have a greater SINR than the minimum required (which is −20 dB, as explained previously), so it serves as an indirect manner to insert the SINR capacity metric into the objective function.

Hence, the objective function is set so that an optimal result should minimize it to zero. Please notice that a weight of 90% to 10% was given in favor of $P_{out(avg)}$, as the greater interest of this study is still to find solutions with maximum coverage probability. This weighting also assures that the number of UAVs suggested by the optimizers must be enough to provide a minimum quantity to associate all users, but also to give in to a greater quantity in order to achieve better outage probabilities if needed.

However, in order to obtain the values for the objective function, we must compute the raw input variables of the positioning of drones until we can obtain SINR, outage, and spectral efficiency values through them. The input variable quantity is the number of maximum drones ($N_{UAV}$) permitted in the simulation multiplied by three, plus an additional number of UAVs suggested by the iteration to solve the problem. Each population iteration will present solutions for drone position data, just as in (Equation 28):(28)$$Pop_{vector}=\left[UAV,x_{UAV1},y_{UAV1},h_{UAV1},x_{UAV2},y_{UAV2},h_{UAV2},\ldots ,x_{UAVn},y_{UAVn},h_{UAVn}\right],$$
where [x,y,h] are the values of each drone in the simulated environment space, and UAV is the number of UAVs that the optimizer has suggested for that iteration. That is, the greater amount of UAVs in the array there are, the greater is the number of inputs, which increases computational cost and time.

Even if the BIC methods do not need the full number of maximum UAVs, the population vector is still kept to the same size since they do not support a dynamic population size along several iterations. So, for instance, if the individual suggests a number of five UAVs, then only those will be considered, and the extra population cells will not be taken into account in the calculations.

Some general equations applied to all algorithms are adaptations to the trigonometrical properties d (distance from user to drone) and θ (elevation angle between user and drone). In order to intertwine the propagation models with the optimization algorithms, values of every population vector are then used to calculate adaptive values of R, d, and θ. This relation is given by Equations (Equation 29)–(Equation 31):(29)$$R_{\left(i,j\right)}=\sqrt{{\left(x_{UAVj}-x_{i}\right)}^{2}+{\left(y_{UAVj}-y_{i}\right)}^{2}},$$
(30)$$\theta_{\left(i,j\right)}=\arctan\frac{h_{UAVj}}{R_{i,j}},$$
(31)$$d_{\left(i,j\right)}=\sqrt{R_{i,j}^{2}+h_{UAVj}^{2}},$$
which all imply that for all j-th drones and i-th users, the difference in their positioning is what creates the variables of distance in the xy-plane (R), total tridimensional distance (d), and elevation angle (θ). So, inside the simulation, an (i by j) matrix of these variables is created in relation to every user (index i) and every UAV (index j).

A flowchart of the simulation and optimization processes is displayed in Figure 5.

## 4. Results

The objective function for all algorithms can be defined as in (Equation 26), where the closer to zero it is, the better the positioning of UAVs will be. It is characterized as a biobjective system, with two variables to be considered.

Inputs for the positioning of drones behave just as explained in Equations (Equation 31) and (Equation 30). They are obligatory for the calculation of distance and elevation angle, which in turn are used to obtain the received power and path loss.

It is worth noticing that all simulations consider only the drones within the array as mobile. Users are kept to fixed locations since the number of users is considerably high and would creased complexity for both simulation coding and results analysis.

There are 24 solutions to be analyzed, considering 3 different optimization techniques (CS, FPA, and GA), 4 different LoRa SF (8, 9, 10, and 11), and 3 distinct path loss models (herein called classical, LDPL, and CI models, respectively).

The lower and upper bounds of the search space were set as a plane of 64 km${}^{2}$ (that is, 8 × 8 km). This is a considerable coverage area, but it is necessary to evaluate how real-life calculations of the LoRa cell range may differ from values provided in [22], present also in Table 2. Additionally, 50 users in this area were generated randomly by normal distribution. This number guarantees that there are users all over the search space whilst still keeping the number of UAVs necessary to provide coverage as not so high. The maximum number of UAVs permitted in all simulations is capped at 10. It must be remembered that there is a limitation of eight users at a time per drone.

In Table 5, a list of relevant variables and constants that define the optimizations for the simulations of LoRaWAN are found. Note that calculations for the empirical models utilize values that are present in Table 3 and Table 4.

Drone height boundaries were set to between 6 and 60 m in order to correlate with data measured at the measurement campaigns (with set heights of [6; 24; 42; 60] m). Therefore, all drones in the simulation can only assume these four values. This makes it easier to compare both path loss models and cell radius capacities.

The transmitted power was kept to around 14 dBm, as many LoRa gateways can support this output to Tx antennas, and works present in the literature have used this value as a reference [42,43]. Additionally, it was the value used in the measurement campaigns. As a matter of reference, [47,48] are a LoRa transceiver and a LoRa gateway, respectively, that can generate up to 22 dBm of Tx output.

All simulations were held in MATLAB© R2021a, on a computer with 16 GB RAM and an AMD Ryzen© 5, 3.6 GHz CPU. Through trial and error to obtain the best results whilst still maintaining a short computational time, 1000 iterations with 25 solutions per iteration were chosen for both LoRa path loss applications. Locking optimizers in a constant iteration cap makes drawing comparisons between them easier.

This section is further divided in the following manner: first, the fitness results and outage probability given by each optimizer will be commented upon for both path loss models. Then, an analysis of the spectral efficiency and cell radius parameters shall be conducted. Lastly, SINR curve plots based on the best UAV positioning values shall be displayed for SF 8 and SF 11.

### 4.1. Fitness and Outage Probability

Table 6 denotes the best fitness values for every algorithm and all simulations made. It also denotes the amount of UAVs that each optimizer has provided as the best, as well as their running time in seconds.

Overall, the cuckoo search presents the best results fitness-wise, but at the cost of a longer running time. Roughly estimating, CS results take twice longer than FPA and around seven times longer than GA. Furthermore, CS and FPA tend to provide lower UAV values to solve the outage problem, which tends to lower fitness. FPA has a satisfactory balance between fitness results and running algorithmic time. GA tends to converge slower towards better solutions iteration-wise, but provides quicker results.

Given that the classical path loss model presents a more “optimistic” approach than realistic measurements, it also provides considerably lower fitness and UAV values. For all LDPL model simulations, the maximum number of 10 UAVs was reached, proving that the channel model based on measured data on suburban Amazon environments actually foresees that coverage area may be considerably lower in real-life scenarios.

On the other hand, the CI model may be an accurate predictor for signal loss behavior next to the UAV station, but it falls short when at larger distances (see Figure 4). So, it also possesses a rather optimistic estimation of range.

It is worth denoting as well that CS and FPA for the classical model application are all capped at seven drones at all simulations, as they recognize that this is the minimum of drones required to associate all users to the network.

The average outage probability of sensors in the simulations is displayed in Figure 6. Results are grouped by BIC method and then separated by SF. Mean outage is given in percentages in the plots, thus keeping results much lower than 1% and successfully achieving optimization across all simulation situations. It is worth denoting that SF 8 tends to show greater outage values due to its lower cell range and sensitivity, while the opposite is true for SF 11.

Results for SF 11 in both path loss models denoted the best outage across all spreading factors, and FPA is, according to Figure 6, the best technique to optimize for lower outage. Fitness values are lower for CS, however, and this may be explained by its lower UAV requirements (both needing fewer drones to optimize and a greater number of users associated). It is also worthy to denote that outage in this range is, approximately, six times greater in the LDPL database compared to the classical one.

### 4.2. Spectral Efficiency and Cell Radius

In Figure 7, the average spectral efficiency for each UE according to the best results of the simulations is shown below. Data are plotted for all simulations, also containing information of the standard error of the mean, which is considerable for this set but has shown to be derisory for outage data.

Between all models, average SEs display no significant difference, only showing slight variations in SF as well. In LDPL data, FPA tends to give lower SE results, while GA and CS are virtually tied for the best SE results. This may indicate that GA should be a viable and swift alternative if the objective is to maximize some kind of spectral efficiency.

Lastly, cell radius estimates for the maximum height (h=60 m) are given in Figure 8. It can be seen that estimates given by (Equation 25) are far below the values given by Semtech, present in Table 2, but for (Equation 24), values are much more proximate. According to [45], cell radius estimates should not include shadowing variables, and yet, values for the range of LoRa ells in measured data are significantly lower than the ideal values. Maximum cell radius for the LDPL model ranges between 914 m (SF 8) and around 1220 m (SF 11) only. Notice that differences tend to become greater as the SF increases.

### 4.3. Sine Contour Plots

Two-dimensional contour plots were compiled for the best results, given by CS, for simulation results of SINR in SF 8 and SF 11. These were chosen because they are both extremes in the measured data. This plot only considers the UAV-to-UAV interference between LoRa gateways, without being tied to a specific channel.

Figure 9 and Figure 10 represent, respectively, the SINR colormap plots of SF 8 and SF 11 (in dB). The white circles denote the position of UEs in the search space, whilst the red dots represent the exact position of UAVs. There is a notable difference in the scale of SINR between the path loss models, where the plot of the classical and CI models tends to undervalue SINR values, while the LDPL may overvalue SINR in shorter distances. However, for both cases, there is a great correlation between drone positioning and the position of the UEs.

Additionally, in both Figure 9 and Figure 10, the difference in coverage for having less or more drones in the simulation can be seen, albeit slightly. There is greater UAV-to-UAV interference when more drones are utilized but, fortunately, not as much to put sensors in the boundaries of cells into outage situations.

Since the simulations shown in Figure 9b and Figure 10b were performed with the same optimizer, path loss model, and transmitted power, it can be attested that drone positioning has only changed slightly between plots. So, the increased range of SF 11 in relation to SF 8 was, in this case, of little difference. As demonstrated in Figure 8, measured data in Amazon suburban environments have a tendency to shorten the expected cell radius considerably, and this may explain the similarities between optimal positioning.

## 5. Conclusions

In this paper, an extensive, simulation-heavy analysis and application of drone-array optimization has taken place. This was made possible by much studying not only optimizations via bioinspired algorithms, but also propagation models for UAV wireless communication systems. The proposal of a simulation environment to determine the optimal positioning of drones in an array system in order to minimize outage probability in MATLAB was accomplished.

The objective of producing fast and accurate responses to the simulation problems proposed herein is achieved by the high computational speed of the bioinspired algorithm techniques. The algorithms are swift, rapid forms of optimization that are meant to be ported into a UAV micro-controller and used on the fly, or even by a remote control station with greater computational power, for faster results. Additionally, the objective of finding a way to relate measured data of LoRa for the slightly forested, suburban Amazon environments into a calculated model has produced more satisfactory and life-like results than purely theoretical, simulation data, as noticed between LoRa simulations in this work.

Metrics such as SINR, spectral efficiency, outage, and cell radius were analyzed for SF 8, 9, 10, and 11. This gives a solid foundation for future works in both suburban and rainforest areas, especially ones with slight-to-medium wooded environments. For the purposes and objectives of the study herein, a low outage was observed in optimizations, despite the reduced range of LoRa cells. In order to compensate for outage, the optimizers suggest a greater quantity of drones, which is one of the differences noted on the SINR plots. Another feature to pay attention to in the SINR colormaps is that there is little concern about outage in cell boundary conditions, as the algorithms seem to comprehend that they should not produce such interference as to leave UEs out of service.

Around all simulations, the best optimized results favor the cuckoo Search algorithm, with flower pollination being a close second, sometimes surpassing the fitness provided by CS but being less taxing in computational cost. The genetic algorithm did not produce the best results in this kind of problem; however, it is lightweight and might be proven useful for cases in which it provides passable accuracy for much less computational time.

However, there are prospects for improvement for future works. For instance, a way to better automatize transmitted power values would be of interest to the literature. For instance, if the code is adapted to minimize the necessary transmitted power for each, which would mean more battery saving for UAVs, transmitters, transceivers, and gateways. Energy efficiency to improve the air time of UAVs is an extensively studied topic throughout the literature, and this could be a welcoming addition.

## Figures and Tables

**Figure 1 sensors-23-06231-f001:**
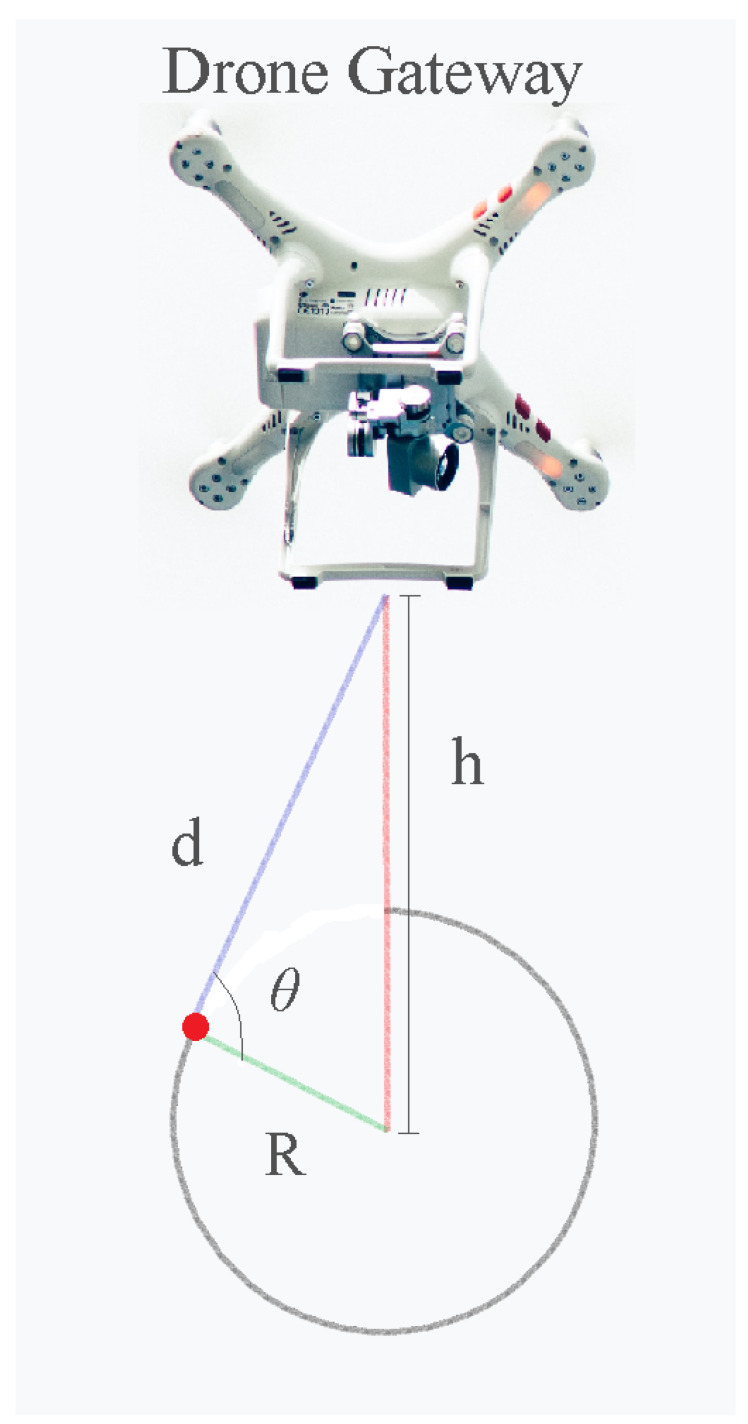
Drone positioning trigonometric variables and schematic.

**Figure 2 sensors-23-06231-f002:**
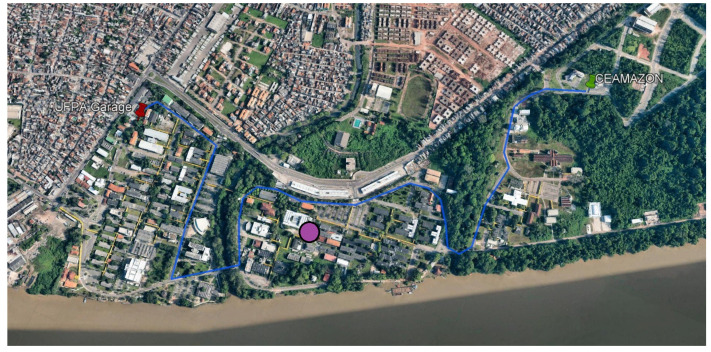
Path taken in all measurement campaigns at UFPA.

**Figure 3 sensors-23-06231-f003:**
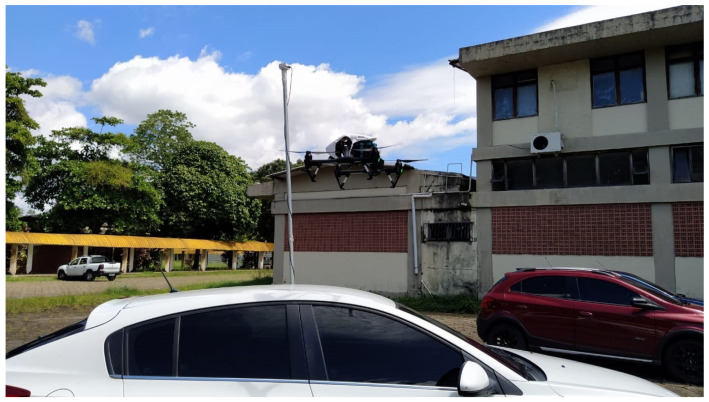
Setup showing the receiving antenna and drone with LoRa gateway.

**Figure 4 sensors-23-06231-f004:**
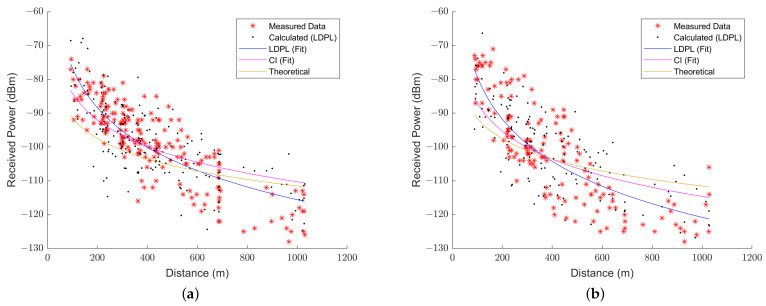
Path loss models, expressed in RSSI, for (**a**) SF 10 at h = 60 m and (**b**) F 11 at h = 24 m.

**Figure 5 sensors-23-06231-f005:**
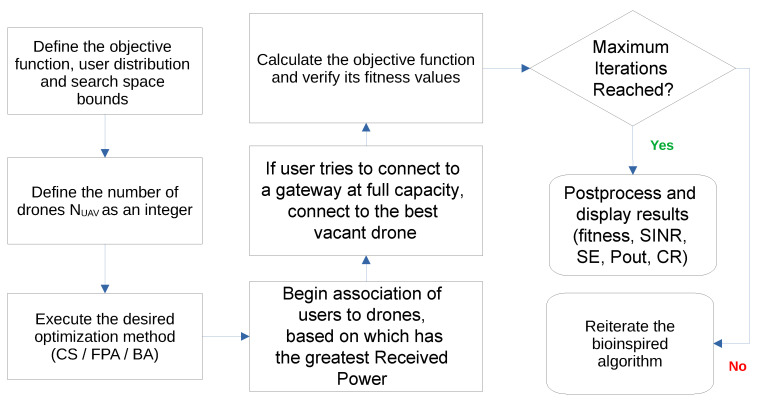
Flowchart of the simulation and optimization processes as made in MATLAB.

**Figure 6 sensors-23-06231-f006:**
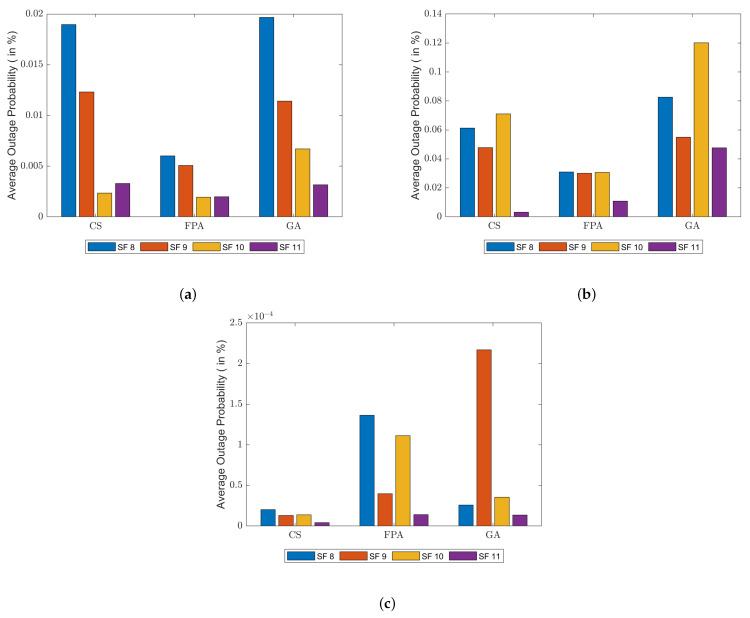
Average outage probability for all simulations: (**a**) classical path loss model; (**b**) LDPL model; (**c**) CI model.

**Figure 7 sensors-23-06231-f007:**
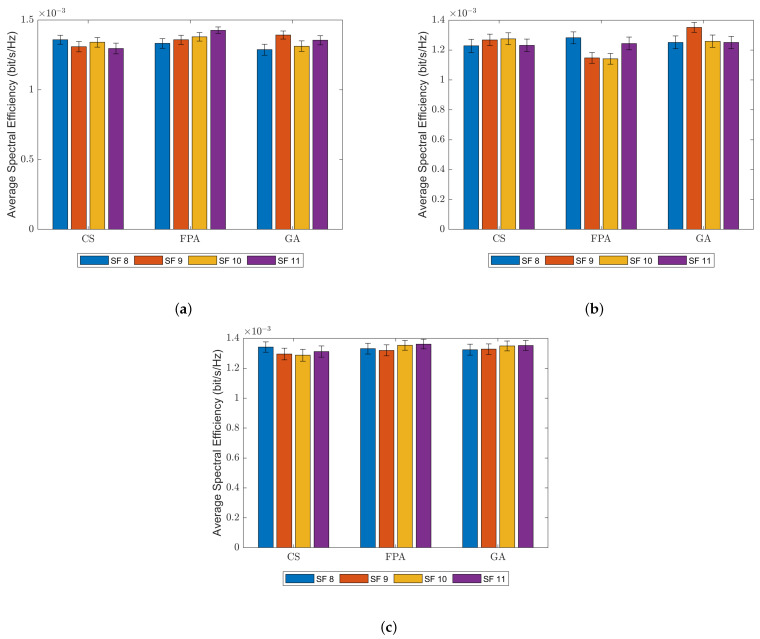
Average spectral efficiency for all simulations (**a**) classical path loss model; (**b**) LDPL model; (**c**) CI model.

**Figure 8 sensors-23-06231-f008:**
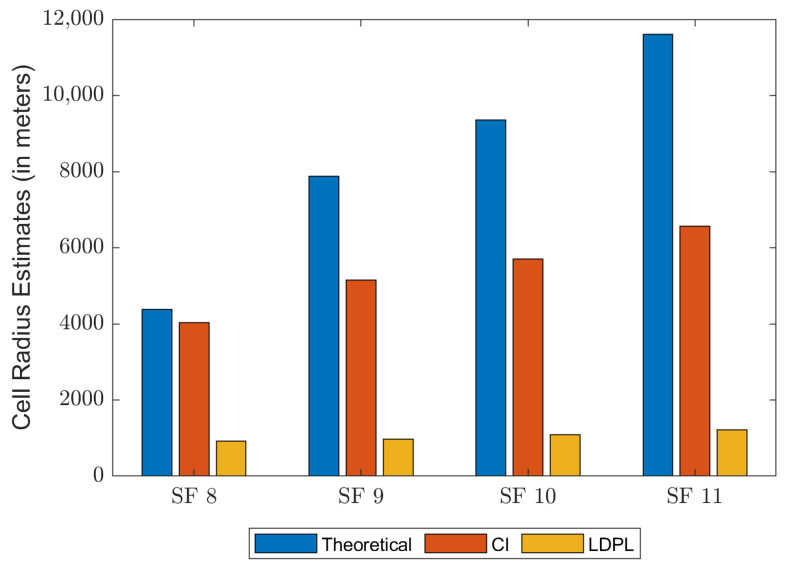
Cell radius estimates for all path loss models.

**Figure 9 sensors-23-06231-f009:**
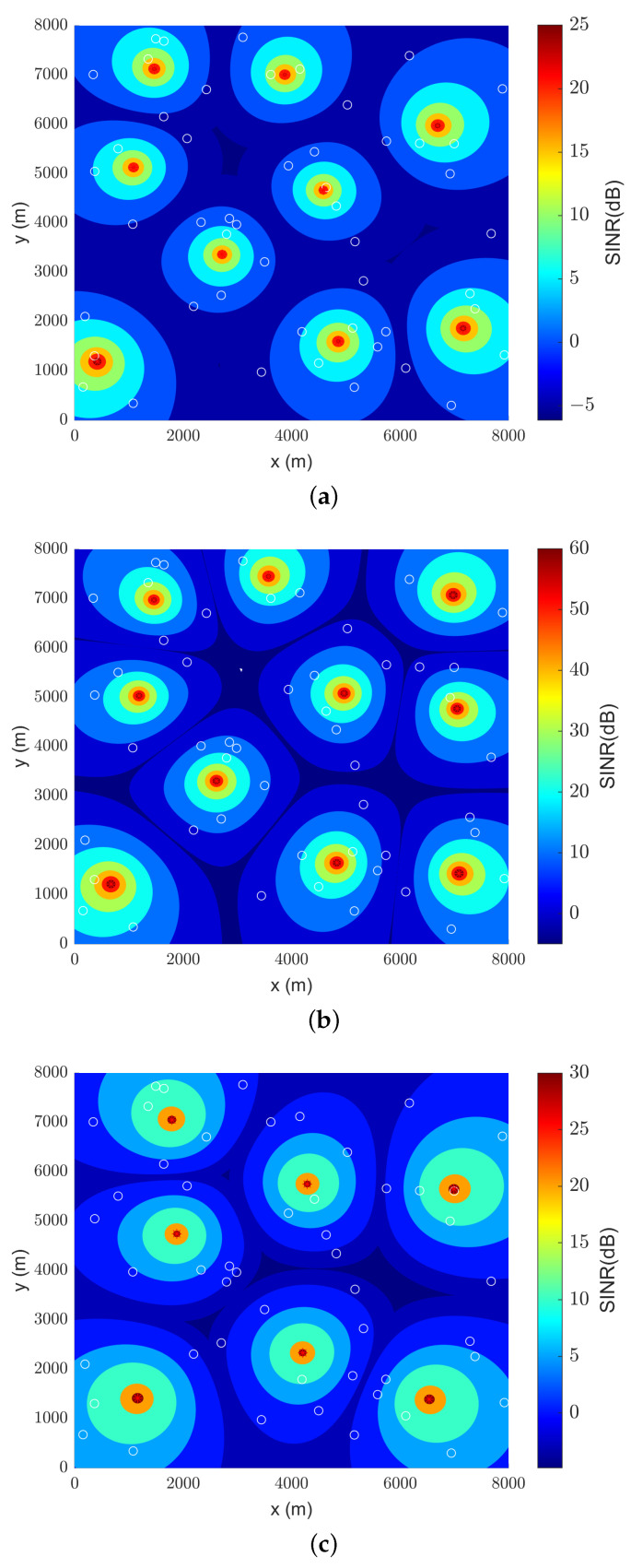
Drone positioning based on SINR values (see color bar) for SF 8: (**a**) classical model; (**b**) LDPL model; (**c**) CI model.

**Figure 10 sensors-23-06231-f010:**
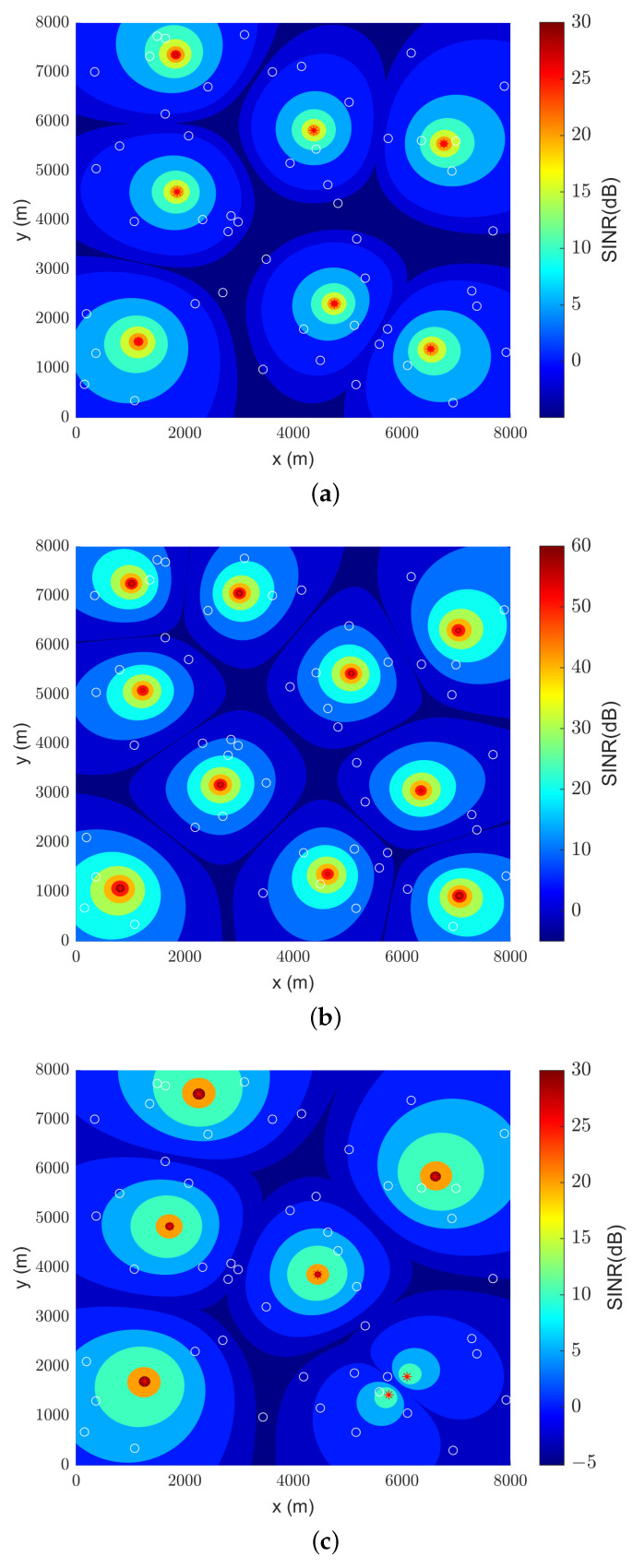
Drone positioning based on SINR values (see color bar) for SF 11: (**a**) classical model; (**b**) LDPL model; (**c**) CI model.

**Table 1 sensors-23-06231-t001:** Contributions of this study in relation to other relevant works.

Work	Multiple UAVs?	LoRa-Related?	Channel Modeling?	Environment	Optimization Method
[3]	–	X	–	Rural	–
[5]	–	X	–	Rural, Suburban	Machine Learning
[6]	–	X	–	Coastal	–
[7]	–	X	–	Rural	–
[8]	X	–	–	Urban	GA, SA
[9]	X	–	–	Urban	PSO
[10]	X	–	–	Urban	PSO
[11]	X	–	–	Urban	PSO
[12]	X	–	–	Urban	PSO, Bayesian
[13] (Survey)	X	–	X	Several	Several
[14] (Survey)	X	–	–	Several	Several
This Study	X	X	X	Suburban, Forested	CS, FPA, GA

**Table 2 sensors-23-06231-t002:** SF reference values of LoRa for f = 915 MHz and BW = 125 kHz. Source: [22,23].

Mode	Bit Rate (kbps)	Sensitivity (dBm)	Δ (dB)	Estimated Value
FSK	1.2	−122	–	N/A
SF 12	0.293	−137	+15	10–12 km
SF 11	0.537	−134.5	+12.5	10 km
SF 10	0.976	−132	+10	8 km
SF 9	1.757	−129	+7	6 km
SF 8	3.125	−126	+4	4 km
SF 7	5.468	−123	+1	2 km
SF 6	9.375	−118	−3	N/A

**Table 3 sensors-23-06231-t003:** Values for the LoRa-LDPL model.

Mode	PLE (n)	σ
SF 11 (h = 6 m)	4.93	9.43
SF 11 (h = 24 m)	4.15	7.95
SF 11 (h = 42 m)	3.92	7.24
SF 11 (h = 60 m)	3.89	6.74
SF 10 (h = 6 m)	4.92	9.93
SF 10 (h = 24 m)	4.29	7.89
SF 10 (h = 42 m)	3.97	7.06
SF 10 (h = 60 m)	3.89	6.47
SF 9 (h = 6 m)	4.98	9.26
SF 9 (h = 24 m)	4.30	6.64
SF 9 (h = 42 m)	4.04	7.15
SF 9 (h = 60 m)	3.92	5.82
SF 8 (h = 6 m)	4.85	9.05
SF 8 (h = 24 m)	4.06	8.61
SF 8 (h = 42 m)	3.85	7.58
SF 8 (h = 60 m)	3.62	6.72

**Table 4 sensors-23-06231-t004:** Values for the LoRa-CI model.

Mode	PLE (n)	σ
SF 11 (h = 6 m)	3.17	9.34
SF 11 (h = 24 m)	2.76	9.43
SF 11 (h = 42 m)	2.63	7.89
SF 11 (h = 60 m)	2.61	7.35
SF 10 (h = 6 m)	3.15	10.12
SF 10 (h = 24 m)	2.82	8.53
SF 10 (h = 42 m)	2.66	7.72
SF 10 (h = 60 m)	2.63	7.41
SF 9 (h = 6 m)	3.17	9.21
SF 9 (h = 24 m)	2.82	6.91
SF 9 (h = 42 m)	2.68	7.14
SF 9 (h = 60 m)	2.61	6.20
SF 8 (h = 6 m)	3.07	9.03
SF 8 (h = 24 m)	2.69	8.68
SF 8 (h = 42 m)	2.59	8.03
SF 8 (h = 60 m)	2.43	7.06

**Table 5 sensors-23-06231-t005:** Variables and constants of the optimizations.

Parameters	Values
Lower bounds (x, y and h)	[0; 0] m
Upper bounds (x, y and h)	[8000; 8000] m
Height bounds (h)	[6, 24, 42, 60] m
Lower and Upper bounds (UAV)	1 to 10 UAVs
Number of iterations (All BIC)	2000
Solutions per iterations (All BIC)	25
$N_{users}$	50
$N_{UAV}$	10
Nest Discard Probability (CS)	0.25
Switch Probability (FPA)	0.3
Crossover Rate (GA)	0.6
Mutation Rate (GA)	0.1
Elitism Rate (GA)	0.1
Transmitted Frequency	915 MHz
Bandwidth	125 kHz
α	9.6
β	0.28
$\zeta_{LOS}$	1 dB
$\zeta_{NLOS}$	20 dB
$G_{t}$ and $G_{r}$	0 dB
Transmitted Power (all UAVs)	14 dBm

**Table 6 sensors-23-06231-t006:** Best fitness outputs.

SF	Algorithm	Best Fitness	UAVs	Run Time (s)
SF=8	CS (Classical)	0.6046	7	185
SF=9	CS (Classical)	0.6000	7	201
SF=10	CS (Classical)	0.5586	7	203
SF=11	CS (Classical)	0.5501	7	197
SF=8	FPA (Classical)	0.6049	7	110
SF=9	FPA (Classical)	0.5544	7	91
SF=10	FPA (Classical)	0.5558	7	94
SF=11	FPA (Classical)	0.5538	7	96
SF=8	GA (Classical)	0.7896	8	28
SF=9	GA (Classical)	0.9001	9	30
SF=10	GA (Classical)	0.7004	8	29
SF=11	GA (Classical)	0.8000	9	27
SF=8	CS (LDPL)	1.7823	10	145
SF=9	CS (LDPL)	2.4969	10	144
SF=10	CS (LDPL)	1.5741	10	145
SF=11	CS (LDPL)	1.2437	10	168
SF=8	FPA (LDPL)	1.8464	10	60
SF=9	FPA (LDPL)	3.1950	10	88
SF=10	FPA (LDPL)	1.9204	10	81
SF=11	FPA (LDPL)	1.2815	10	84
SF=8	GA (LDPL)	4.4031	10	27
SF=9	GA (LDPL)	6.4865	10	29
SF=10	GA (LDPL)	4.0611	10	27
SF=11	GA (LDPL)	2.4539	10	26
SF=8	CS (CI)	0.6002	7	159
SF=9	CS (CI)	0.6002	7	163
SF=10	CS (CI)	0.5534	7	167
SF=11	CS (CI)	0.5504	7	164
SF=8	FPA (CI)	0.5649	7	75
SF=9	FPA (CI)	0.5553	7	68
SF=10	FPA (CI)	0.5766	7	72
SF=11	FPA (CI)	0.5555	7	71
SF=8	GA (CI)	0.8966	10	28
SF=9	GA (CI)	0.5758	7	27
SF=10	GA (CI)	0.9011	10	26
SF=11	GA (CI)	0.7567	9	29

## Data Availability

Data on the inputs and outputs of this study, as well as the codes that generated them, are available at https://drive.google.com/drive/folders/1ORq-5LkzXcIiqqeorFSi1Djj6z58Gu7C?usp=share_link. Last access: 29 June 2023.

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
