# Peer review of "Intelligent Drone Positioning via BIC Optimization for Maximizing LPWAN Coverage and Capacity in Suburban Amazon Environments"

_sensors, 2023, doi:10.3390/s23136231_

Round 1

Reviewer 1 Report

The article "Intelligent Drones Positioning via BIC Optimization for Maximizing LPWAN Coverage and Capacity in Suburban Amazon Environments" is weel organized and presented. 

Need minor corrections to process

1) The existing possible methods related comparative analysis not presented  to get complete insight of novelty in the work

2) The conclusion is too heavy and it can be shortened with focus on the outcome

Can be improved

Author Response

RESPONSE TO THE REVIEWERS
We would like to thank all reviewers for their suggested remarks. We shall answer them below, one by one, as ordered
in the review reports.
Some major considerations before the answers are to be addressed. The empirical model proposed in this work,
based on the Log-Distance Path Loss model, has been modified and, as a result, the simulations had to be redone. As
said by Reviewer 2, the fitting of data was, in fact, unconvincing. We have modified it to provide different equations
for every SF and all 4 different heights, as well as take into consideration the transmitted power (Pt = 14 dBm). This
results in 16 different curve fittings, from previously only 4.
The height bounds in the simulations have changed to adapt to this new empirical model as well. The height of
UAVs in simulation is now fixed at 6, 24, 42, and 60 meters, and simulations for the empirical model now use the
appropriate equation for each height. We have also increased the number of iterations for the BICs to 2000 from 1000,
in order to better optimization results after these changes. Since the maximum execution time is still under 4 minutes,
we believe there are no issues with this.
Therefore, we kindly remind you to review our results and let us know if there are any inconsistencies.

REVIEWER 1
Thank you for the suggested changes to the conclusion. We shall answer your remark below:
Reviewer Comment 1.1 — The existing possible methods related to comparative analysis are not presented to get complete
insight of novelty in the work.
Reply: Thank you for the insight. We have separated a part of Related Works to show our contributions,
and have elaborated a table that compares the methods and contributions of other papers to our own (lines
126 to 137, Table 1).
Reviewer Comment 1.2 — The conclusion is too heavy and it can be shortened with a focus on the outcome.
Reply: We have streamlined the conclusion. Please let us know if there are any other issues.

Reviewer 2 Report

1. The main concern I have about the paper is with respect to the fitting degree between the proposed propagation model and the measured data you obtained. Pages 4-5, equations 3-5, the path loss for LOS and NLOS connections in Mozaffari. M et al. (2015) didn’t apply to the scenario the paper mentioned, and yet you didn’t modify it according to the environment that the empirical model should be employed. As well as Page 6, equation 10, the log-distance path loss model in El Chall. R et al. (2019) was conducted under the environment in Lebanon, a place quite different from the Amazon Forest. Since you didn’t further revise it based on the specific circumstance the empirical model was designated to, I am unconvinced about the fitting degree between the empirical model and the measured data. In Figure. 4, it is obvious that the measured data is much denser than calculated data, and become increasingly apparent in Figure. 5, while all SFs are shown in the same graph. Please adjust the empirical model or append some parameters for the purpose that the calculated data can be more accurate and closer to the measured one.

2. My related concern about the analysis is the design of the objective function. I found that you already announced the equation of SINR previously, and I wonder why you didn’t introduce it into your objective function? I understand that the number of UAVs and the outage probability is quite important in coverage problem. However, the QoS, represented by SINR, should be taken into account since you mentioned capacity in your title. The result you noted on lines 567-570, the topology of drones may be changed considerably with the revision of the objective function and thus enrich the results.

3. Section 3.3, line 438, you mentioned UAV is the optimal quantity calculated by the optimizers. Due to the fact that you only claimed the usage of BIC optimization was to maximum the coverage of the drones-array, I assume that the quantity of the drones may be optimized by other methods you didn’t mention. I suggest you to refine your expression about the optimal quantity of the drones.

4. Can you please arrange the figures in a compact way? For example, Figure. 1 and Figure. 3 can be reduced and arranged in the same row, because they are both with small amount of information and mainly for illustration. As for the Figure. 4 and Figure. 5, there is no need to let figures occupy too much space in a page. Likewise, for the figures in section 4, you can reduce and plot subgraphs in the same row so that the page can be more concise.

5. Section 3.1, line 166, eqs? Eqs? Equations? Please unify the expression of the equations throughout the paper.

6. Section 3.3, line 401, what does UE mean? Is it an acronym for User Equipment? Please give the full name on first time.

7. Section 3.3, line 408, why you choose the tolerable value around -20 dBm? It seems you forget to finish your citation because of the existence of brackets.

8. Section 3.3, line 417, why you use the symbol PL(i,j) in the equation 21 but PL(d)  in the main body of the paper? Please unify them in case of misleading the readers.

9. Section 4.1, line 525 and line 532, please add ‘Figure’ before ‘7’.

Minor editing of English language required

Author Response

RESPONSE TO THE REVIEWERS
We would like to thank all reviewers for their suggested remarks. We shall answer them below, one by one, as ordered in the review reports.
Some major considerations before the answers are to be addressed. The empirical model proposed in this work, based on the Log-Distance Path Loss model, has been modified and, as a result, the simulations had to be redone. As
said by Reviewer 2, the fitting of data was, in fact, unconvincing. We have modified it to provide different equations for every SF and all 4 different heights, as well as take into consideration the transmitted power (Pt = 14 dBm). This results in 16 different curve fittings, from previously only 4.
The height bounds in the simulations have changed to adapt to this new empirical model as well. The height of UAVs in the simulation is now fixed at 6, 24, 42, and 60 meters, and simulations for the empirical model now use the appropriate equation for each height. We have also increased the number of iterations for the BICs to 2000 from 1000, in order to better optimization results after these changes. Since the maximum execution time is still under 4 minutes,
we believe there are no issues with this.
Therefore, we kindly remind you to review our results and let us know if there are any inconsistencies.

REVIEWER 2
Reviewer Comment 2.1 — The main concern I have about the paper is with respect to the fitting degree between the proposed propagation model and the measured data you obtained. Pages 4-5, equations 3-5, the path loss for LOS and NLOS connections in Mozaffari. M et al. (2015) didn’t apply to the scenario the paper mentioned, and yet you didn’t modify it according to the environment in that the empirical model should be employed. As well as Page 6, equation 10, the long-distance path loss model in El Chall. R et al. (2019) were conducted in the environment of Lebanon, a place quite different from the Amazon Forest. Since you didn’t further revise it based on the specific circumstance the empirical model was designated to, I am unconvinced about the fitting degree between the empirical model and the measured data. In Figure. 4, it is obvious that the measured data is much denser than the calculated data, and this becomes increasingly apparent in Figure. 5, while all SFs are shown in the same graph. Please adjust the empirical model or append some parameters for the purpose that the calculated data can be more accurate and closer to the measured one.
Reply: Thank you for the remarks. We have concluded that our fitting for the empirical model was suboptimal, as it did not consider the UAV height on-air, and this considerably alters the estimation of the model.
So, instead of 4 Equations, we have adapted the model to have an Equation for each height and SF variation, totaling 16 Equations, each with a different fitting.
Values for the new 16 fittings are observed in Table 3, and Figure 4 denotes two examples of said fittings.

As for the validity of the Log-Distance model: it is well understood that it is a generic and adaptable model that relies on measured data to validate itself to the specific environment it wishes to model. Our work does not use the values of path loss exponent and standard deviation from El Chall. R et al. (2019) - this article is only used as an example of application. We have put another reference showing the use of the Log-Distance model in a dense forest environment, which is relevant to our study (see the paragraph of Lines 222 to 228).
Regarding the adaptation of the model by Mozaffari. M et al. (2015), the parametric values of α = 9.6 and β = 0.28 reflect a suburban environment, so we believe there are no other adaptations to be done. Also, it is a model to be used as a comparison reference in this work, due to its more ”optimistic” results in relation to the empirical model proposed in the manuscript.
In summary, we believe that the fittings and the path loss models are better justified now, and the results obtained in the simulations are more accurate. Please let us know if there are any other issues on the matter.
Reviewer Comment 2.2 — My related concern about the analysis is the design of the objective function. I found that you already announced the equation of SINR previously, and I wonder why you didn’t introduce it into your objective function. I understand that the number of UAVs and the outage probability is quite important in coverage problem.
However, the QoS, represented by SINR, should be taken into account since you mentioned capacity in your title.
The result you noted on lines 567-570, the topology of drones may be changed considerably with the revision of the objective function and thus enrich the results.
Reply: The SINR is already indirectly inserted into the objective function in the form of the variable N associated.
Network users can only be considered associated with LoRa channels if they meet SINR requirements, which is an SINR above -20 dBm. Some works in the literature support this requirement for LoRaWAN (see Line 424). Therefore, the UAV Requirements parameter of the objective function deals with capacity by optimizing how many drones are required and also optimizing as many users as possible to an acceptable SINR. We have inserted more information on this matter in Lines 457 to 460.
Reviewer Comment 2.3 — Section 3.3, line 438, you mentioned UAV is the optimal quantity calculated by the optimizers. Due to the fact that you only claimed the usage of BIC optimization was to maximize the coverage of the
drones-array, I assume that the quantity of the drones may be optimized by other methods you didn’t mention. I suggest you refine your expression about the optimal quantity of the drones.
Reply: It does provide a number of drones only in the objective function. The results, after the revisions, corroborate our findings that the use of this UAV Requirements variable keeps the number of drones to a minimum value to satisfy user association necessities, with CS and FPA returning 7 UAVs necessary to provide for 50 users in lower outage situations. Also, for the empirical model simulations in which an outage is much greater, it has adapted itself to provide more drones to cover the outage deficit, with the optimizers suggesting 10 UAVs.
We have put an explanation for this in Lines 461 to 466, Section 3.3., and Lines  45 to 552 of the Results.
Reviewer Comment 2.4 — Can you please arrange the figures in a compact way? For example, Figure. 1 and Figure.
3 can be reduced and arranged in the same row, because they are both with small amount of information and mainly for illustration. As for the Figure. 4 and Figure. 5, there is no need to let figures occupy too much space on a page.
Likewise, for the figures in section 4, you can reduce and plot subgraphs in the same row so that the page can be more concise.
Reply: We thank you for the suggestions. We are concerned that Figures 2 and 3 are too wide to put into a single row, so we have kept them as is. Figure 1 has been reduced in size, however.
As for Figures 4 and 5, they have been arranged into a single row and are now Figures 4a and 4b. Figures 7 and 8 have their subfigures in the same row, and are Figure 6 and Figure 7 in the new revision, respectively.
However, we have chosen to maintain the formatting of the SINR contour plots (now Figures 9 and 10), given that they may benefit from greater size to facilitate the ascertainment of results by the readers.

Reviewer Comment 2.5 — Section 3.1, line 166, eqs.? Eqs? Equations? Please unify the expression of the equations throughout the paper.
Reply: Thank you for the correction. We have modified all expressions to  Equation” and ”Equations”, in full.
Figures and Tables are also referenced in full.
Reviewer Comment 2.6 — Section 3.3, line 401, what does UE mean? Is it an acronym for User Equipment? Please give the full name the first time.
Reply: It is, indeed. The full name at first appearance is now given (Line 417).
Reviewer Comment 2.7 — Section 3.3, line 408, why did you choose the tolerable value around -20 dBm? It seems you forget to finish your citation because of the existence of brackets.
Reply: Yes, the references are now inserted correctly (Line 424).
Reviewer Comment 2.8 — Section 3.3, line 417, why do you use the symbol PL(i,j) in equation 21 but PL(d) in the main body of the paper? Please unify them in case of misleading the readers.
Reply: Thank you for noticing this, we have now corrected it to PL(i, j) in both text and equation forms.
Reviewer Comment 2.9 — Section 4.1, line 525, and line 532, please add ‘Figure’ before ‘7’.
Reply: It is now corrected.

Reviewer 3 Report

This paper proposed the propagation channel model of radio waves by considering the presence of vegetation. The following articles discuss how vegetation affects radio wave propagation. Therefore, this paper does not have enough novelty for the publication of this paper on the MDPI Sensors.

J. Ko et al., "28 GHz millimeter-wave measurements and models for signal attenuation in vegetated areas," 2017 11th European Conference on Antennas and Propagation (EUCAP), 2017, pp. 1808-1812, doi: 10.23919/EuCAP.2017.7928520.   

P. Zhang, B. Yang, C. Yi, H. Wang and X. You, "Measurement-Based 5G Millimeter-Wave Propagation Characterization in Vegetated Suburban Macrocell Environments," in IEEE Transactions on Antennas and Propagation, vol. 68, no. 7, pp. 5556-5567, July 2020, doi: 10.1109/TAP.2020.2975365.

may be acceptable

Author Response

We would like to thank all reviewers for their suggested remarks. We shall answer them below, one by one, as ordered in the review reports.
Some major considerations before the answers are to be addressed. The empirical model proposed in this work, based on the Log-Distance Path Loss model, has been modified and, as a result, the simulations had to be redone. As
said by Reviewer 2, the fitting of data was, in fact, unconvincing. We have modified it to provide different equations for every SF and all 4 different heights, as well as take into consideration the transmitted power (Pt = 14 dBm). This
results in 16 different curve fittings, from previously only 4.
The height bounds in the simulations have changed to adapt to this new empirical model as well. The height of UAVs in simulation is now fixed at 6, 24, 42, and 60 meters, and simulations for the empirical model now use the
appropriate equation for each height. We have also increased the number of iterations for the BICs to 2000 from 1000, in order to better optimization results after these changes. Since the maximum execution time is still under 4 minutes,
we believe there are no issues with this.
Therefore, we kindly remind you to review our results and let us know if there are any inconsistencies.

REVIEWER 3
Reviewer Comment 3.1 — This paper proposed the propagation channel model of radio waves by considering the presence of vegetation. The following articles discuss how vegetation affects radio wave propagation. Therefore, this paper does not have enough novelty for the publication of this paper on the MDPI Sensors.
Reply: With all due respect, this statement only covers one of the many contributions of our paper.
As stated in our paper (Lines 128 to 135): ”No papers were found on the propagation of LoRa, or LPWANs
in general, in forested environments. Also, no study has conciliated the usage of UAV arrays and their positioning optimizations with channel modeling in forested environments. And finally, there are not many studies on the usage of UAVs for signal propagation in Amazon rainforest regions - let alone on drone positioning and coverage optimization”.
Hence, there is the study of LoRa propagation, coverage, and capacity in Amazon suburban environments, the utilization of UAV arrays to provide LoRa signal to sensors on the ground, and the simulations that utilize Bioinspired Computing for optimal deployment of a sensor network via drone arrays. These are all major contributions of the work that should not be taken separately.
Furthermore, the references you have provided are made in the millimeter-wave range for 5G applications, which is not the case for Wireless Sensor Networks that utilize much lower frequencies such as LoRa (915 MHz, in this case). So, we believe it is difficult to affirm that such studies could be applied for our
measurements and simulations.
We have also inserted a Table comparing our contributions to other works in the literature (Table 1).
We thank you for your concern, and if you have any suggestions let us know.

Round 2

Reviewer 3 Report

The authors presented the parameter estimation results for the specific wireless propagation channel. Although these results may be effective, the effectiveness of this result is for limited readers. As a scientific and engineering paper, the authors present new methodologies. Unfortunately, this method relies on ordinary radio wave propagation theory. Therefore, the novelty of this paper is not enough as an MDPI paper. 

Author Response

Dear Reviewer,
We are delighted to receive your contributions, and we believe that our article has the potential to be published and referenced in the Sensors journal, as we have achieved success in all the areas addressed, both in the employed model and in the measurements.
Notwithstanding the fact that the work was developed in a region of the Amazon rainforest, which in itself represents a significant challenge given the unique conditions we have here, the relevance of successful studies like ours, embedded in a reality known only by the local population who are familiar with the challenges faced, enables us to seek publications in renowned journals such as Sensors.
Furthermore, for the state-of-the-art section, we present results for mixed urban environments and extremely challenging signal propagation and maintenance of wireless systems in densely forested Amazonian areas. It is worth highlighting that we have tree canopies reaching over a hundred meters in relation to the ground, in addition to constant torrential rains.
The results presented in our work will strongly contribute to further research, as previously mentioned, and the promising findings we are presenting will undoubtedly lead to citations.
In this second round of revisions, and in response to your valuable comments, we have added another path loss model to the adjustments and simulations, namely the nearby free space or CI model. This will provide greater validity to the manuscript's findings. Please refer to the tracked changes in the PDF file for more information.
Figures 4a and 4b have been modified to accommodate the adjustment of all three propagation models. Figures 6, 7, 8, 9, and 10 have also been altered to showcase the results of the simulations conducted using the CI model.
Additionally, if there are any errors or suggestions, please let us know.
Sincerely,
Miércio Neto – Author
REVIEWER 3
Reviewer Comment 0.1 — The authors presented the parameter estimation results for the specific wireless propagation
channel. Although these results may be effective, the effectiveness of this result is for limited readers. As a scientific
and engineering paper, the authors present new methodologies. Unfortunately, this method relies on ordinary radio
wave propagation theory. Therefore, the novelty of this paper is not enough as an MDPI paper.
Reply: Papers about LoRa and IoT utilizing wave propagation theory and channel modelling have been published as soon as this year - e.g. Robles-Enciso et al. (2023). Supramongkonset et al. (2021) and Onykiienko et al. (2022) are other recent examples. This means there is a strong demand to aggregate environment-specific channel modelling and UAV-positioning solutions to LoRaWAN in the literature.
Furthermore, the empirical channel modelling found in the methodology of the paper has been divided into each height and each SF, in order to provide better fitting of results and depth of results to analyze. The need to define channel models and parameters that can be used in the Amazon region also gives the study a solid context.
Nevertheless, we have included the CI propagation model to our simulations and fitting of data, to provide further robustness to the manuscript’s findings. The CI model is present in many recent works in the literature, such as Park et al. (2023), Caso et al. (2021) and aforementioned Robles-Enciso et al. (2023) and Onykiienko et al. (2022).
Based on the arguments provided above, we believe that the manuscript is a novel and valid contribution in the areas of channel modelling in wooded environments and UAV implementation of IoT systems.
REFERENCES
Caso, G., Alay, O., De Nardis, L., Brunstrom, A., Neri, M., and Di Benedetto, M.-G. (2021). Empirical models for nb-iot path loss in an urban scenario. IEEE Internet of Things Journal, 8(17):13774–13788.
Onykiienko, Y., Popovych, P., Yaroshenko, R., Mitsukova, A., Beldyagina, A., and Makarenko, Y. (2022). Using rssi data for lora network path loss modeling. In 2022 IEEE 41st International Conference on Electronics and Nanotechnology (ELNANO), pages 576–580. IEEE.
Park, J., Jeon, H.-B., Cho, J., and Chae, C.-B. (2023). Measurement-based close-in path loss modeling with diffraction for rural long-distance communications. IEEE Wireless Communications Letters.
Robles-Enciso, R., Morales-Aragon, I. P., Serna-Sabater, A., Mart ́ ́ınez-Ingles, M. T., Mateo-Aroca, A., Molina-Garcia- Pardo, J.-M., and Juan-Llacer, L. (2023). Lora, zigbee and 5g propagation and transmission performance in an indoor environment at 868 mhz. Sensors, 23(6):3283.
Supramongkonset, J., Duangsuwan, S., Maw, M. M., and Promwong, S. (2021). Empirical path loss channel characterization based on air-to-air ground reflection channel modeling for uav-enabled wireless communications. Wireless
Communications and Mobile Computing, 2021:1–10
